# Extensive T-Cell Profiling Following SARS-CoV-2 mRNA Vaccination in Multiple Sclerosis Patients Treated with DMTs

**DOI:** 10.3390/pathogens14030235

**Published:** 2025-02-27

**Authors:** Hannah Solchenberger, Marcus Odendahl, Dirk Schriefer, Undine Proschmann, Georges Katoul al Rahbani, Tjalf Ziemssen, Katja Akgün

**Affiliations:** 1Center of Clinical Neuroscience, Department of Neurology, Carl Gustav Carus University Hospital, Technical University Dresden, 01307 Dresden, Germany; hannah.solchenberger@ukdd.de (H.S.); dirk.schriefer@ukdd.de (D.S.); undine.proschmann@ukdd.de (U.P.); georges.katoulalrahbani@ukdd.de (G.K.a.R.); tjalf.ziemssen@ukdd.de (T.Z.); 2Medical Faculty Carl Gustav Carus, Technical University Dresden, Experimental Transfusion Medicine, 01307 Dresden, Germany; m.odendahl@blutspende.de; 3Institute for Transfusion Medicine Dresden, German Red Cross Blood Donation Service North-East, 01307 Dresden, Germany

**Keywords:** multiple sclerosis, mRNA vaccines, SARS-CoV-2, disease-modifying therapy, anti-CD20 treatment, sphingosine-1-phosphate receptor modulation, T-cell profiling

## Abstract

Disease-modifying therapies (DMTs) are known to impact cellular and humoral immune response in persons with multiple sclerosis (pwMS). In this study, we performed in-depth SARS-CoV-2-specific T-cell profiling using flow cytometry. T-cell immunity in pwMS with or without DMTs was evaluated before a first SARS-CoV-2 messenger ribonucleic acid (mRNA) vaccination and at one-, two- and six-month follow-up. T-cell stimulation without SARS-CoV-2-specific antigens was used as a control. T-cell response was compared to B-cell response by evaluating SARS-CoV-2-specific antibodies. We observed an upregulation of specific subpopulations of SARS-CoV-2 spike-specific CD4^+^ T cells. Thus, our results demonstrate the induction of a broad and distinct CD4^+^ T-cell response in pwMS even on anti-CD20 treatment and sphingosine-1-phosphate receptor modulation after SARS-CoV-2 mRNA vaccination. This was particularly seen in CD4^+high^ and CD4^+^CD154^+^ T cells. Our results do not support the induction of a CD8^+^ T-cell immune response. While humoral immune response was impaired in pwMS during ocrelizumab and fingolimod treatment, there was evidence of a compensatory upregulation of subpopulations of SARS-CoV-2-specific CD4^+^ T cells at low levels of seroconversion in pwMS. In conclusion, our results provide important insights into the mechanisms of the adaptive immune response in pwMS following SARS-CoV-2 mRNA vaccination.

## 1. Introduction

Multiple sclerosis (MS) is a chronic inflammatory disease of the central nervous system that affects 2.9 million people worldwide, with its prevalence still on the rise [1,2]. Treatment strategies for this disease, which is not yet curable, consist of relapse treatment, disease-modifying therapy (DMT) and symptomatic therapy [3,4,5]. The development of messenger ribonucleic acid (mRNA) vaccines has progressed rapidly during the coronavirus disease 19 (COVID-19) pandemic. Since DMTs model immune function, the question arises as to what extent a positive immune response to mRNA vaccination against SARS-CoV-2 is possible in people with MS (pwMS) treated with DMTs.

The efficacy of SARS-CoV-2 mRNA vaccination in inducing cellular and humoral immune responses in immunocompetent patients has been well established [6,7,8,9]. In contrast, the picture in pwMS, especially when treated with certain DMTs, is less clear. Previous data showed a reduced humoral immune response after mRNA vaccination against SARS-CoV-2 during therapy with anti-CD20 antibodies or sphingosine-1-phosphate receptor (S1PR) modulators [10,11,12,13]. Interestingly, several studies found that during anti-CD20 therapy antibody levels increased in correlation with the time since the last infusion and the extent of B-cell reconstitution during vaccination [10,14,15,16]. With S1PR modulators, some authors have suggested that a low absolute lymphocyte count is the cause of a lack of immune response [10,17]. In addition, data from our team showed that the measurement of humoral immune responses is not sufficient to characterize virus-specific immune responses. Therefore, the detection of T-cell immunity is also helpful in assessing the efficacy of SARS-CoV-2 mRNA vaccination [14,18]. We and others have previously demonstrated robust antigen-specific CD4^+^ and CD8^+^ T-cell responses after SARS-CoV-2 mRNA vaccination under anti-CD20 therapy, even in the presence of weak antibody responses [12,14,16,19]. In addition, our previous data support the hypothesis that in the absence of B cells and antibodies, there is a compensatory increase in T cell-mediated responses to SARS-CoV-2 mRNA vaccination [15]. Several studies also found a reduced cellular immune response following SARS-CoV-2 mRNA vaccination with S1PR modulation [11,19,20,21,22]. In contrast, another study found that despite significantly reduced anti-receptor-binding domain (RBD) levels of the spike protein during treatment with fingolimod, a preserved neutralizing response was observed. Additionally, the cellular immune response was comparable to untreated pwMS and healthy donors [23].

In the present study, we performed a longitudinal systematic evaluation of SARS-CoV-2-specific T-cell immunity, including analysis of activation markers, differentiation markers, and cytokines after mRNA vaccination in pwMS treated with the anti-CD20 antibody ocrelizumab or the S1PR modulator fingolimod compared to a control group. In addition, we aimed to directly compare SARS-CoV-2-specific T- with B-cell immunity and to establish a flow cytometry-based assay particularly suitable for the detection of T-cell immunity in immunosuppressed therapy constellations.

## 2. Materials and Methods

Study design and patients: This prospective cohort study was conducted at the MS Center of Carl Gustav Carus University Hospital, Dresden, Germany. A total of 47 pwMS were recruited between March 2021 and January 2022. The cohort was divided into three groups based on the current DMT. One group consisted of 15 pwMS receiving ocrelizumab, and another group consisted of 15 pwMS receiving fingolimod. The control group consisted of 17 pwMS who were either not receiving immunomodulatory therapy or were receiving therapy with natalizumab or alemtuzumab (last cycle at least two years ago). Due to the limited availability of pwMS not on medication, pwMS being treated with alemtuzumab and natalizumab were also included in the control group, as no reduction in vaccine efficacy was expected with these drugs based on the mode of action and data from vaccinations other than SARS-CoV-2 [24,25,26]. In addition, there were not enough pwMS who had received potent immunosuppressive DMTs such as cladribine early before SARS-CoV-2 mRNA vaccination, so we did not include these pwMS in the study. All patients were vaccinated with mRNA vaccines approved in Germany. They received either 30 µg of the BNT162b2 vaccine (BioNTech SE, Mainz, Germany) or 100 µg of the mRNA-1273 vaccine (Moderna, Cambridge, MA, USA) for their first vaccination. The SARS-CoV-2 mRNA vaccines contain nucleoside-modified RNA, directing the synthesis of the antigen that elicits an immune response, namely the spike protein of SARS-CoV-2 [27]. Exclusion criteria for enrollment in the study were SARS-CoV-2 infection confirmed by polymerase chain reaction (PCR) or rapid antigen testing prior to the first blood sample collection, vaccination against SARS-CoV-2 with a non-mRNA-based vaccine, and age less than 18 years.

Blood sampling: Heparinized and serum blood samples were collected at four defined time points for the immunological biobank of the Neuroimmunology Laboratory: before the first vaccination with a SARS-CoV-2 mRNA vaccine (baseline = BL) and one (M1), two (M2), and six months (M6) after the first vaccination. Peripheral blood mononuclear cells (PBMCs) were isolated from heparinized blood samples using Pancoll (Pan-Biotech GmbH, Aidenbach, Germany) for density gradient centrifugation. Cryopreservation was required to analyze blood samples collected at different times. PBMCs were resuspended in a freezing medium consisting of 10% dimethyl sulfoxide (Merck KGaA, Darmstadt, Germany) and fetal calf serum (Fisher Scientific GmbH, Schwerte, Germany). Samples were stored in a nitrogen tank at a minimum temperature of −170 °C until further processing.

Immune cell phenotyping of SARS-CoV-2-specific T cells by fluorescence-activated cell sorting (FACS): Frozen PBMCs were thawed, washed, and resuspended in culture medium consisting of RPMI 1640 (Fisher Scientific GmbH), 5% human AB serum (c.c.pro, Oberdorla, Germany), 200 mM L-glutamine (Fisher Scientific GmbH), 10,000 U/mL penicillin, and 10,000 μg/mL streptomycin (Fisher Scientific GmbH) before the cell count was adjusted to 2 × 10^6^ cells/mL and cells were incubated for 1 h (37 °C, 5% CO_2_). The monoclonal antibody anti-human CD28 (Becton Dickinson GmbH, Heidelberg, Germany) diluted to 1.5% and PepTivator^®^ peptide pools (Miltenyi Biotec B.V. & Co. KG, Bergisch Gladbach, Germany) consisting of lyophilized peptides (60 nmol/peptide) diluted to 0.238% were then added. According to the manufacturer’s instructions, the peptide pool for the spike (S) protein covers its immunodominant regions, and the peptide pool for the S1 domain of the S protein (S1) covers the N-terminal domain of the S-protein. The peptide pools were added separately to stimulate the T cells. As a positive control, 3.85% CytoStim^TM^ human (Miltenyi Biotec B.V. & Co. KG) was added instead of peptide pools, and stimulation with anti-human CD28 (Becton Dickinson GmbH) alone was used as a negative control. After 1 h incubation, 1% protein transport inhibitor GolgiPlug^TM^ containing brefeldin A (Becton Dickinson GmbH) was added to facilitate the detection of intracellular cytokines. This was followed by a further incubation of 4 h, after which cell viability was determined using Zombie Green^TM^ (BioLegend, Koblenz, Germany). Cells were stored in FACS buffer consisting of phosphate-buffered saline (Merck KGaA, Darmstadt, Germany), 0.5% inactivated fetal bovine serum (Fisher Scientific GmbH), and 10% sodium azide for one night at 4 °C. The cells were then fixed with 4% paraformaldehyde (Merck KGaA), followed by permeabilization of the cell membrane with 0.1% saponin to enable intracellular cytokine detection. Subsequently, the cells were incubated with fluorescence-labeled antibodies. For analysis of surface antigens, cells were incubated with anti-cluster of differentiation (CD)-3, anti-CD4, anti-CD8, anti-HLA-DR, anti-CD38, anti-CD45RA and anti-CCR7 monoclonal antibodies (Becton Dickinson GmbH). For additional characterization of intracellular markers, cells were incubated with anti-CD154, anti-IFN-γ, anti-IL-2, and anti-TNF monoclonal antibodies (Becton Dickinson GmbH). After completion of staining, cells were analyzed on a BD LSRFortessa™ Cell Analyzer (Beckton Dickinson GmbH), and relative percentages of T-cell subsets were determined for further evaluation (Appendix A).

Detection of SARS-CoV-2-specific antibodies: To determine humoral immunity, an electrochemiluminescence immunoassay (ECLIA) was performed on an automated Cobas e 801 system (Roche, Basel, Switzerland) in a certified laboratory affiliated with the Institute of Transfusion Medicine, Dresden, Germany, under standardized conditions. Immunoglobulin G (IgG) antibodies directed against the RBD of the S protein of SARS-CoV-2 were measured in blood serum. The seropositivity cut-off was defined as 0.8 U/mL according to the manufacturer’s instructions. The lower limit of detection was 0.43 U/mL. Values < 0.43 U/mL were reported as 0.2 U/mL. The upper detection limit was 25,000 U/mL. The unit U/mL corresponds to the unit defined by the WHO as binding antibody units (BAU)/mL and did not require conversion [28].

Statistical analysis: Sociodemographic and disease-specific data were collected. Continuous variables are described as means and standard deviations (SD), whereas categorical variables are described as absolute (*n*) and relative frequencies (%). Analyses were performed for the entire study sample and for individual subgroups. The distribution of each relative T-cell population and anti-SARS-CoV-2 RBD IgG titers was examined using histograms, quantile–quantile plots, and the Shapiro–Wilk test. To answer the key research questions, generalized linear mixed models (GLMMs) were used for longitudinal analysis of each individual T-cell population and the anti-SARS-CoV-2 RBD IgG titers. In addition to the time point (BL, M1, M2, M6), the patient group, the interaction of time point and patient group, the anti-SARS-CoV-2 RBD IgG titers, and the interaction of anti-SARS-CoV-2 RBD IgG titers and patient group were included in the models for the analysis of T-cell populations. A negative control was considered in the study. The GLMM for analysis of anti-SARS-CoV-2 RBD IgG titers included the time point, the patient group and the interaction of time point and patient group. For normally distributed data, a GLMM with normal distribution and identity link function was applied. For right-skewed variables, a GLMM with gamma distribution and log link function was used. Left-skewed variables were mirror-transformed (reflected) to have a right-skewed distribution and then analyzed in the same way as the right-skewed data. Bonferroni correction was used to account for type 1 statistical error for pairwise (between patient groups) and simple (time points after vaccination to BL, seronegative IgG to seropositive) comparisons. Outcomes for all models are reported as model estimates (mean and 95% confidence interval). *p* values were defined as significant as follows: * *p* < 0.05, ** *p* < 0.01, *** *p* < 0.001. Flow cytometry analysis was performed using FlowJo 10.8.1 (Becton, Dickinson & Company, Ashland, OR, USA). IBM^®^ SPSS Statistics 29.0.1.0 (IBM Corporation, Armonk, NY, USA) was used for statistical analysis and the data were visualized with GraphPad Prism 9.00 (GraphPad Software, Boston, MA, USA).

## 3. Results

### 3.1. Patient Characteristics

The study cohort comprised 47 pwMS divided into three groups: 17 in the control group, 15 receiving ocrelizumab, and 15 receiving fingolimod. The average age was 45.60 ± 11.89 years and 35 (74.5%) pwMS were female, while 12 (25.5%) were male (Table 1). In sum, 38 (80.9%) were diagnosed with relapsing–remitting MS (RRMS). Four (8.5%) had a secondary progressive and five (10.6%) a primary progressive disease course. Disease duration was similar among the three groups, whereas treatment duration appeared to be shorter in the ocrelizumab group compared to pwMS treated with other DMTs (Table 1). EDSS before BL did not differ significantly among the groups, but patients in the control group tended to have higher EDSS scores compared to the other two groups (Table 1). There were no differences among the groups in the time intervals from the first vaccination to the time points (Table 1).

In addition, the cohort was characterized based on the types of vaccines administered and the number of vaccinations at each time point (Table 2).

### 3.2. SARS-CoV-2-Specific T-Cell Response

The primary objective of this study was to evaluate the SARS-CoV-2-specific T-cell response after SARS-CoV-2 mRNA vaccination. For this purpose, percentages of spike-specific T cells (S) and T cells specific for the S1 domain of the spike protein (S1) were analyzed before a first SARS-CoV-2 mRNA vaccination and at one-, two- and six-month follow-up in the three patient groups and compared to each other. Due to its mode of action, fingolimod induces a depletion of lymphocytes in the peripheral blood, resulting in lymphocyte distribution that is very different from that observed in pwMS in the ocrelizumab and in the control group [29,30]. This difference is particularly relevant for the following analysis of relative T-cell subsets upon SARS-CoV-2-specific stimulation.

The relative percentage of SARS-CoV-2-specific CD4^+^ T cells did not change significantly over time in the entire study cohort, with only the control and ocrelizumab group showing a trend towards an increasing percentage of CD4^+^ T cells after vaccination (Figure 1A,B). However, all patient groups differed significantly in their percentage of CD4^+^ T cells (*p*(S) < 0.001; *p*(S1) < 0.001), with the ocrelizumab group having the highest levels, followed by the control group, and then the fingolimod group with the lowest levels (Figure 1A,B, Appendix A). Relative CD4^+^ T cells in the overall cohort and within each patient group did not differ with respect to anti-SARS-CoV-2 RBD IgG serostatus (Table 3).

Flow cytometric analysis identified two distinct SARS-CoV-2-specific CD4^+^ T-cell populations based on their fluorescence intensity—CD4^+high^ and CD4^+low^ T cells—which were analyzed in addition to CD4^+^ T cells (Appendix A). A previous study also demonstrated that CD4^+^ and CD8^+^ T cells in general can be distinguished into subpopulations based on their antigen expression, showing either high or low fluorescence intensity after staining with monoclonal anti-CD4 or anti-CD8 antibodies [31].

Relative CD4^+high^ T cells showed a significant increase in the entire study cohort after SARS-CoV-2 mRNA vaccination compared to BL (*p*(S) = 0.002; *p*(S1) = 0.003) (Figure 1C,D). Similarly to CD4^+^ T cells, the number of CD4^+high^ T cells in the patient groups differed significantly (*p*(S) < 0.001; *p*(S1) < 0.001), with the highest percentage of CD4^+high^ T cells in the ocrelizumab group, followed by the control and fingolimod group (Figure 1C,D, Appendix A). In the overall study cohort, there was a significantly higher percentage of S-specific CD4^+high^ T cells with positive anti-SARS-CoV-2 RBD IgG than with negative IgG (*p* = 0.049), but not for the S1-specific T cells (Table 3). Relative CD4^+high^ T cells within individual patient groups did not differ significantly based on serostatus (Table 3).

For CD4^+low^ T-cell percentage, only a decreasing trend was observed in the entire study cohort at two- and six-month follow-up (Figure 1E,F). The percentage of SARS-CoV-2-specific CD4^+low^ T cells differed significantly between patient groups (*p*(S) < 0.001; *p*(S1) < 0.001), with the fingolimod group having the highest levels, followed by similar percentages in the control and ocrelizumab groups (Figure 1E,F, Appendix A). In the overall cohort, the percentage of CD4^+low^ T cells did not differ based on anti-SARS-CoV-2 IgG serostatus. However, analysis of individual patient groups revealed a significant effect in the ocrelizumab group, where seronegative individuals had a significantly higher percentage of CD4^+low^ T cells compared to seropositive individuals (*p*(S) = 0.001; *p*(S1) < 0.001) (Table 3).

Regarding the percentage of SARS-CoV-2-specific CD8^+^ T cells, there were no significant differences over time (Figure 1G,H). There were significant differences between patient groups (*p*(S) < 0.001; *p*(S1) < 0.001), with the fingolimod group having a significantly higher percentage of CD8^+^ T cells compared to the other two groups (Figure 1G,H, Appendix A). The percentage of CD8^+^ T cells did not differ based on anti-SARS-CoV-2 RBD IgG in the overall study cohort or in the individual patient groups (Table 3).

The percentage of S-specific CD4^+^CD154^+^ T cells in the overall study cohort showed a significant increase over time (*p* < 0.001), particularly at one and two months after the first vaccination, with a slight decrease at six months. Despite this decrease, the percentage remained significantly higher than at BL (Figure 2A). This effect was not significant for S1-specific T cells (Figure 2B). Relative CD4^+^CD154^+^ T cells also differed significantly between patient groups (*p*(S) < 0.001; *p*(S1) < 0.001), with the ocrelizumab group showing significantly lower levels compared to the other two groups (Figure 2A,B, Appendix A). As with CD4^+high^ T cells, analysis of the entire cohort revealed that pwMS with anti-SARS-CoV-2 negative RBD IgG had a significantly higher percentage of S-specific CD4^+^CD154^+^ T cells than those with anti-SARS-CoV-2 positive RBD IgG (*p* = 0.001). This effect was not significant for S1-specific T cells (Table 3). Also, the percentage of CD4^+^CD154^+^ T cells in the individual patient groups did not differ significantly based on serostatus (Table 3).

The percentages of SARS-CoV-2-specific CD4^+^IFN-y^+^, CD4^+^IL-2^+^ and CD4^+^TNF-α^+^ T cells did not differ significantly over time in the overall study cohort. However, there was a trend of increasing percentages of T-cell subsets in all three patient groups at the one-month follow-up, followed by a decreasing trend at the six-month follow-up (Figure 2C–H). None of these CD4^+^ T-cell populations showed significant differences in relative percentages of T cells based on anti-SARS-CoV-2 RBD IgG serostatus in the overall cohort or within each patient group (Table 3).

### 3.3. SARS-CoV-2-Specific T-Cell Response—Activation and Differentiation Markers

For the activation markers CD38 and HLA-DR, significant differences over time were observed only in relative percentages of SARS-CoV-2-specific CD4^+^CD38^−^HLA-DR^+^ T cells (*p*(S) < 0.001; *p*(S1) < 0.001), which decreased significantly six months after the first vaccination compared to BL in the entire study cohort (Appendix A). Other T-cell populations of activation markers showed no differences over time (Appendix A). The fingolimod group had a significantly higher percentage of CD4^+^CD38^−^HLA-DR^+^ T cells compared to the other two groups (*p*(S) < 0.001; *p*(S1) < 0.001), while it had significantly lower percentages of CD4^+^CD38^+^HLA-DR^−^, CD4^+^CD38^−^HLA-DR^−^ and S1-specific CD4^+^CD38^+^HLA-DR^+^ T cells (*p*(S) < 0.001; *p*(S1) < 0.001) (Appendix A). In the overall study cohort, no differences were observed based on anti-SARS-CoV-2 RBD IgG serostatus for any activation marker T-cell populations. However, in the ocrelizumab group, pwMS with negative anti-SARS-CoV-2 RBD IgG had significantly higher percentages of CD4^+^CD38^+^HLA-DR^+^ T cells compared to those with positive IgG (*p*(S) = 0.008; *p*(S1) = 0.018), with a similar effect for S1-specific CD4^+^CD38^−^HLA-DR^−^ T cells (*p* = 0.018) (Appendix A).

Concerning the differentiation markers CCR7 and CD45RA, there were no significant differences over time or in relation to anti-SARS-CoV-2 RBD IgG serostatus in relative SARS-CoV-2-specific naïve CD4^+^ T cells (CD4^+^CD45RA^+^CCR7^+^), CD4^+^ effector memory T cells (TEMs; CD4^+^CD4RA^−^CCR7^−^), or CD4^+^ terminal effector memory T cells (TEMRAs; CD4^+^CD45RA^+^CCR7^−^) in the entire study cohort or in each individual group. However, S1-specific CD4^+^ central memory T cells (TCMs; CD4^+^CD45RA^−^CCR7^+^) showed a significant decrease over time (*p* = 0.015), with significantly lower percentages of T cells after the first vaccination compared to BL (Appendix A). Regarding percentages of T cells in the patient groups, significant differences were observed for naïve CD4^+^ T cells (*p*(S) = 0.030; *p*(S1) = 0.015), CD4^+^ TCMs (*p*(S) < 0.001; *p*(S1) < 0.001) and CD4^+^ TEMs (*p*(S) < 0.001; *p*(S1) < 0.001). Naïve CD4^+^ T cells and CD4^+^ TCMs were significantly reduced in the fingolimod group, while CD4^+^ TEMs were significantly increased in this group (Appendix A). Additionally, in the entire cohort, pwMS with positive anti-SARS-CoV-2 RBD IgG titers had significantly higher percentages of S1-specific CD4^+^ TCMs than those with negative titers (*p* = 0.047). Within the individual patient groups, seropositive pwMS in the control group had significantly higher percentages of CD4^+^ TCMs compared to seronegative pwMS (*p* = 0.020) (Appendix A).

There were no significant differences in percentages of T cells over time for SARS-CoV-2-specific CD8^+^CD154^+^, CD8^+^IFN-y^+^, CD8^+^IL-2^+^, and CD8^+^TNF-α^+^ T cells in the entire study cohort (Appendix A). With respect to patient groups, only the percentage of CD8^+^IL-2^+^ T cells showed significant differences, with the ocrelizumab group having significantly higher percentages of T cells than the other two groups (*p*(S) < 0.001; *p*(S1) = 0.001) (Appendix A). In the overall cohort, no differences were observed in percentages of CD8^+^IFN-y^+^, CD8^+^IL-2^+^ and CD8^+^TNF-α^+^ T cells based on anti-SARS-CoV-2 RBD IgG serostatus. However, for S1-specific CD8^+^CD154^+^ T cells, a significantly higher percentage of T cells was observed in seropositive individuals compared to seronegative ones (*p* < 0.035) (Appendix A). Regarding patient groups, a significantly higher percentage of S-specific CD8^+^IL-2^+^ T cells was found in seronegative individuals in the ocrelizumab group compared to seropositive ones (*p* = 0.014), whereas no significant effect was observed for the S1-specific T cells (Appendix A).

With respect to the activation markers HLA-DR and CD38 on CD8^+^ T cells, no significant differences in relative T cells over time were observed in the whole study cohort (Appendix A). There were significantly more CD8^+^CD38^−^HLA-DR^+^ T cells in the ocrelizumab group than in the fingolimod group (*p*(S) < 0.001; *p*(S1) < 0.001) and significantly more CD8^−^CD38^−^HLA-DR^−^ T cells in the fingolimod group than in the control group (*p*(S) < 0.001; *p*(S1) < 0.001) (Appendix A). No significant differences in percentages of T cells were found in the overall cohort or within individual patient groups based on the serostatus of anti-SARS-CoV-2 RBD IgG (unpublished data).

Concerning the differentiation markers CCR7 and CD45RA, no significant changes were observed over time for SARS-CoV-2-specific CD8^+^ TCMs, CD8^+^ TEMs, or CD8^+^ TEMRAs in the overall study cohort (Appendix A). The percentage of SARS-CoV-2-specific CD8^+^ naïve T cells showed a significant decrease over time in the fingolimod group (*p*(S) = 0.002; *p*(S1) = 0.001), with significantly lower percentage of T cells at one- (*p*(S) = 0.006; *p*(S1) = 0.004), two- (*p*(S) < 0.001; *p*(S1) < 0.001) and six-month follow-up (*p*(S) < 0.001; *p*(S1) < 0.001) compared to BL. No significant difference in CD8^+^ naïve T cells was observed over time in the overall study cohort (Appendix A). Additionally, all groups differed significantly regarding their percentages of T cells of differentiation markers, with CD8^+^ naïve, CD8^+^ TCMs, CD8^+^ TEMRAs (*p*(S) < 0.001; *p*(S1) < 0.001), and CD8^+^ TEMs (*p*(S) = 0.006; *p*(S1) = 0.012) showing significant variations. The percentages of CD8^+^ naïve T cells and CD8^+^ TCMs were significantly reduced in the fingolimod group, while CD8^+^TEMs and CD8^+^TEMRAs were significantly increased in this group (Appendix A). No significant differences in total cell counts based on anti-SARS-CoV-2 RBD IgG serostatus were observed in the overall cohort or within the individual patient groups (unpublished data).

### 3.4. SARS-CoV-2-Specific B-Cell Response

The total anti-SARS-CoV-2 RBD IgG titers showed a significant increase over time compared to BL (*p* < 0.001) (Figure 3A). Six months after the first vaccination, antibody titers had decreased but were still significantly higher than those at BL (Figure 3A). In addition, significant differences in total anti-SARS-CoV-2 RBD IgG levels were observed between the patient groups when all time points were considered together (*p* < 0.001). The control group presented significantly higher levels of IgG than the ocrelizumab and the fingolimod group. In addition, the fingolimod group also had higher IgG levels than the ocrelizumab group, which had the lowest levels (Figure 3B).

In accordance with this, seropositivity for anti-SARS-CoV-2 RBD IgG differed among the three patient groups. In the control group, 82.4% (14/17) of pwMS had seroconverted one month after the first vaccination and 100% (17/17) two and six months after the first vaccination. In contrast to that, only 26.7% (4/15) of pwMS receiving ocrelizumab had seroconverted one month after the first vaccination. After two months, only 20% (3/15) and after six months, only 13.3% (2/15) of these pwMS were anti-SARS-CoV-2 RBD IgG seropositive. Under fingolimod treatment 46.7% (7/15) of pwMS had seroconverted one month after the first vaccination, 60% (9/15) after two months, and 66.7% (10/15) after six months (Table 4).

## 4. Discussion

The development of mRNA vaccines has progressed rapidly during the COVID-19 pandemic, and these vaccines have been shown to induce robust SARS-CoV-2-specific T- and B-cell immunity in healthy populations [6,7,8]. However, data on the efficacy and safety of these vaccines in pwMS remain not so clear. Consequently, it is crucial to monitor and define immunocompetence following vaccination. The aim of our study was to systematically evaluate SARS-CoV-2-specific T-cell immunity after SARS-CoV-2 mRNA vaccination in pwMS treated with ocrelizumab and fingolimod and in a control group and to compare T- with B-cell immunity. Previous publications predominantly relied on SARS-CoV-2-specific IFN-γ to quantify cellular immunity after SARS-CoV-2 mRNA vaccination [11,14,20,21,23,32]. Our study aimed to establish a flow cytometry-based assay with a comprehensive panel of activation and differentiation markers, along with cytokines, specifically designed to evaluate antigen-specific T-cell immunity in immunosuppressed patients.

Percentages of SARS-CoV-2-specific CD4^+^ T cells did not show a significant increase after SARS-CoV-2 mRNA vaccination. However, when only SARS-CoV-2-specific CD4^+high^ T cells were analyzed, a significant increase was observed at all time points across the entire study cohort following SARS-CoV-2 mRNA vaccination compared to BL. This provides the first clear indication of positive cellular immunity after SARS-CoV-2 mRNA vaccination. Furthermore, the significantly higher percentage of S-specific CD4^+high^ T cells in pwMS with negative anti-SARS-CoV-2 RBD IgG compared to those with positive IgG is striking. This may indicate a compensatory upregulation of the CD4^+^ T-cell response in the absence of a B-cell response. We have previously demonstrated an increased S-specific IFN-γ CD4^+^ and CD8^+^ T-cell response after SARS-CoV-2 mRNA vaccination in pwMS on anti-CD20 therapy compared to untreated pwMS, further supporting the hypothesis of compensatory upregulation of T cell-mediated response in the absence of a B-cell response [15]. In peripheral blood, the percentage of SARS-CoV-2-specific CD4^+^ T cells and percentage of CD4^+high^ T cells were significantly reduced in the fingolimod group compared to the two other groups. This aligns with the mode of action of fingolimod, which inhibits lymphocyte egress—particularly CD4^+^ naïve T cells and CD4^+^ TCMs—from secondary lymphatic organs into peripheral blood circulation through S1PR modulation [33,34,35]. Other studies have reported reduced absolute CD4^+^ T-cell counts after SARS-CoV-2 mRNA vaccination, as well as reduced relative numbers of S-specific CD3^+^CD4^+^ T cells before and after SARS-CoV-2 mRNA vaccination in pwMS treated with fingolimod [20,22]. In contrast, the percentage of SARS-CoV-2-specific CD4^+low^ T cells showed no significant changes following SARS-CoV-2 mRNA vaccination, and the significantly increased percentage of CD4^+low^ T cells in the fingolimod group compared to the two other groups is striking. One possible hypothesis for these dynamics is that SARS-CoV-2-specific CD4^+^ T cells are upregulated after SARS-CoV-2 mRNA vaccination, leading to an increase in CD4^+high^ T cells over time. This upregulation may not be possible in pwMS treated with fingolimod, resulting in an increased number of CD4^+low^ T cells. Consequently, treatment with fingolimod may contribute to a diminished vaccine response or complicate its assessment due to the redistribution of lymphocytes, as the measurement of T cells in lymph nodes is not feasible. Additionally, the different levels of CD4 expression may be attributed to varying affinity for the stimulating antigen among the patient groups. To gain a deeper understanding, we therefore conducted an analysis of additional functional markers on immune cells. The percentage of SARS-CoV-2-specific CD8^+^ T cells was significantly increased in the fingolimod group compared to the two other groups at all time points. This is in line with a previous study that reported an increased relative number of S-specific CD3^+^CD8^+^ T cells in pwMS treated with fingolimod before and after SARS-CoV-2 mRNA vaccination [22]. Moreover, it highlights the distinct migratory behavior of CD8^+^ T cells compared to CD4^+^ T follicular helper cells. However, we observed no differences in relative CD8^+^ T-cell percentage over time, in contrast to another study that found a significant decrease in absolute CD3^+^CD8^+^ T-cell counts after SARS-CoV-2 vaccination in pwMS on fingolimod compared to pwMS on natalizumab and healthy controls [20].

We observed a significant increase in relative S-specific CD4^+^CD154^+^ T cells (CD40L) after SARS-CoV-2 mRNA vaccination across the entire study cohort. CD154 is a costimulatory surface marker expressed on activated CD4^+^ T cells and promotes B-cell activation for an effective immunoglobulin class switch, B-cell differentiation, and affinity maturation [9,36,37]. Consequently, CD154 has been used in several studies as an activation marker for SARS-CoV-2-specific CD4^+^ T cells following SARS-CoV-2 mRNA vaccination [6,7,9,16,22]. Meyer-Arndt et al. reported a significant increase in S antigen-specific CD4^+^CD154^+^4-1BB^+^ T cells after the first SARS-CoV-2 mRNA vaccination in pwMS treated with ocrelizumab and natalizumab and in those without DMT. In contrast, a significantly reduced or absent S antigen-specific T-cell response was observed under fingolimod treatment [22]. Apostolidis et al. described a comparable significant increase in S antigen-specific CD4^+^CD154^+^CD200^+^ T cells after SARS-CoV-2 mRNA vaccination in pwMS treated with ocrelizumab and in healthy controls [16]. However, in our study, the percentage of SARS-CoV-2-specific CD4^+^CD154^+^ T cells was significantly lower in the ocrelizumab group compared to the two other groups. This difference may be due to the general reduced expression of CD154 during anti-CD20 therapy [36,38,39]. Consistent with findings on S-specific CD4^+high^ T cells, the higher percentage of S-specific CD4^+^CD154^+^ T cells in seronegative pwMS compared to seropositive pwMS with respect to SARS-CoV-2 RBD IgG further supports the idea of a compensatory upregulation of SARS-CoV-2-specific CD4^+^ T-cell response in the absence of a B-cell response.

Our results did not show a significant increase in the percentages of cytokines (IFN-y, IL-2, TNF-α), typically secreted by TH1-T cells in SARS-CoV-2-specific CD4^+^ T cells after SARS-CoV-2 mRNA vaccination. However, trends towards increased T-cell percentages for these cytokines were observed at one- and two-month follow-up. Other authors claim a polyfunctionality of S-specific CD4^+^ T cells with comparable release of IL-2, TNF-α, and IFN-γ between healthy controls and pwMS undergoing anti-CD20 therapy [12]. Similarly, another study found comparable cytokine responses among healthy controls, untreated pwMS, and pwMS treated with natalizumab, S1PR modulators, and anti-CD20 therapy [40]. Based on our findings, it is plausible that with a larger sample and more pwMS per group, the hypothesis of a significant increase in SARS-CoV-2-specific CD4^+^ T-cell cytokine production could be confirmed.

Our study does not confirm the induction of activation markers on CD4^+^ or CD8^+^T cells. Nonetheless, other studies have reported pronounced upregulation of HLA-DR and CD38 on CD4^+^ and CD8^+^ T cells during SARS-CoV-2 infection and in non-seroconverted patients under anti-CD20 therapy compared to seroconverted patients [41,42]. Notably, the fingolimod group showed a significantly higher percentage of CD4^+^CD38^−^HLA-DR^+^ T cells compared to the other two patient groups, but significantly reduced percentages of the other three T-cell subsets (CD4^+^CD38^+^HLA-DR^+^, CD4^+^CD38^+^HLA-DR^−^, CD4^+^CD38^−^HLA-DR^−^ T cells). This may be explained by the reduced peripheral CD4^+^ T-cell count caused by fingolimod treatment, potentially indirectly diminishing the detection of activation markers in peripheral blood.

Concerning differentiation markers on CD4^+^ T cells, only the percentage of S1-specific CD4^+^ TCMs decreased after vaccination across the study cohort, with no changes observed in SARS-CoV-2-specific CD4^+^ naïve T cells, CD4^+^ TEMs, or CD4^+^ TEMRAs over time. Other studies have reported increased CD4^+^ TCM and CD4^+^ TEM counts postvaccination in healthy controls and pwMS on anti-CD20 therapy [6,7,12,16,43,44]. The percentages of SARS-CoV-2-specific CD4^+^ TCMs were significantly higher in seropositive compared to seronegative pwMS across the entire cohort, most likely driven by significantly elevated CD4^+^ TCMs in seropositive individuals in the control group compared to seronegative individuals and markedly reduced CD4^+^ TCMs in the fingolimod group. In addition to CD4^+^ TCMs, the percentage of CD4^+^ naïve T cells was also significantly reduced in the fingolimod group, while CD4^+^ TEMs were significantly increased compared to the two other groups. The reduction in naïve CD4^+^ T cells and CD4^+^ TCMs is likely linked to their expression of the homing receptor CCR7, which promotes their retention in lymph nodes by fingolimod [45].

We observed no differences in the release of CD154 and cytokines by SARS-CoV-2-specific CD8^+^ T cells over time, suggesting a dominant SARS-CoV-2-specific CD4^+^ T-cell response. Consistently with this, other studies have reported significantly higher production of TH-1 cytokines by S-specific CD4^+^ T cells compared to CD8^+^ T cells in healthy individuals, including the elderly [46,47]. However, other authors found comparable cytokine production by S-specific CD4^+^ and CD8^+^ T cells after SARS-CoV-2 mRNA vaccination in pwMS without DMT or those treated with anti-CD20 therapy, alemtuzumab, fingolimod, and healthy controls [23]. Interestingly, percentages of SARS-CoV-2-specific CD8^+^IL-2^+^ T cells were significantly elevated in the ocrelizumab group compared to the two other groups. This might indicate an increased production of SARS-CoV-2-specific CD8^+^ cytokines during anti-CD20 therapy although no significant differences were found over time.

Regarding differentiation markers, we observed a decrease in relative SARS-CoV-2-specific CD8^+^ naïve T cells after vaccination in the fingolimod group and no differences in the percentages of SARS-CoV-2-specific CD8^+^ TCMs, CD8^+^ TEMs, or CD8^+^ TEMRAs. This suggests that these cell populations were not significantly affected by the SARS-CoV-2 mRNA vaccine. Moreover, CD8^+^ TCMs were significantly reduced, while CD8^+^ TEMs and CD8^+^ TEMRAs were significantly increased in the fingolimod group, reflecting the typical effects of S1PR modulators on T cells.

With respect to B-cell immunity, our data showed an intact humoral immune response in pwMS, both untreated and treated with natalizumab or alemtuzumab longer before, while it was impaired in those treated with ocrelizumab or fingolimod. The seroconversion rate in pwMS treated with fingolimod and ocrelizumab was significantly lower, and their anti-SARS-CoV-2 RBD IgG titers reached significantly lower levels compared to the control group. These findings align with previous studies, demonstrating a significantly lower SARS-CoV-2 antibody response in pwMS treated with anti-CD20 therapy or S1PR modulation [10,20,22,40,48].

Our study has several limitations. First, the monocentric study design limits the generalizability of our results, as the specific characteristics of our center and personal and environmental factors involved may influence the findings. We did not validate our results in an independent cohort. The small sample and consequently the relatively limited size of the compared groups reduced statistical power and made it challenging to draw definitive conclusions, particularly regarding the small differences observed between SARS-CoV-2-specific T-cell populations. Some non-significant trends in T-cell responses after mRNA vaccination suggest that certain statistical differences might not have been detected. Additionally, the control group was not homogeneous due to the limited number of pwMS without DMT, which required the inclusion of pwMS on treatment. The number of time points and the intervals between them may not have been optimal for fully characterizing the immune response, as a shorter interval and more time points could have provided further insight into T- and B-cell responses. Furthermore, there was no evaluation of the immune response after the second mRNA vaccine, so any effects on cell populations due to this second immunization cannot be excluded. Another limitation is the lack of separation in the analysis of cells from the two licensed mRNA vaccines (BNT162b2 and mRNA-1273), which may have differing effects. Detection of prior SARS-CoV-2 infection and subsequent exclusion of subjects was based solely on a positive PCR test, without the additional measurement of anti-N-specific antibodies, as has been done by many others [10,20,30,42,48,49,50]. Consequently, asymptomatic SARS-CoV-2 infections may not have been detected. Finally, when using peptide pools to quantify T-cell immunity, it should be noted that peptide concentration, peptide length, and the range of overlap of peptides in the peptide pool may influence the T-cell response [51,52].

## 5. Conclusions

In summary, our results demonstrate the induction of a broad SARS-CoV-2-specific CD4^+^ T-cell response in pwMS under anti-CD20 therapy and S1PR modulation following SARS-CoV-2 mRNA vaccination. Thus, pwMS in immunosuppressed therapy constellations achieve at least partial protection against severe COVID-19, although our data could not confirm the induction of a SARS-CoV-2-specific CD8^+^ T-cell response. Additionally, the humoral immune response following SARS-CoV-2 mRNA vaccination is impaired during anti-CD20 therapy and S1PR modulation. Interestingly, the findings indicate a compensatory upregulation of SARS-CoV-2-specific CD4^+^ T cells in the absence of or with low levels of seroconversion in pwMS. The development of a comprehensive flow cytometry assay suitable for the quantification of antigen-specific cellular immunity in other immunosuppressed therapy constellations succeeded. This approach emphasizes that specific T-cell populations may be more suitable for monitoring vaccination responses, and antigen-specific analyses are likely more accurate than non-specific cellular monitoring. However, it is crucial to note that our results cannot be directly applied to all DMTs, as cellular responses may vary depending on the mode of action and immunological processes cannot be uniformly evaluated and consequently measured in peripheral blood. Further research is needed to address the impact of booster vaccinations and breakthrough infections on long-term SARS-CoV-2-specific cellular and humoral immunity. It can be concluded that cellular immunity should be considered as important as humoral immunity when assessing immune responses after SARS-CoV-2 mRNA vaccination. This provides the opportunity to optimize vaccination strategies and vaccination success in pwMS in the best possible way.

## Figures and Tables

**Figure 1 pathogens-14-00235-f001:**
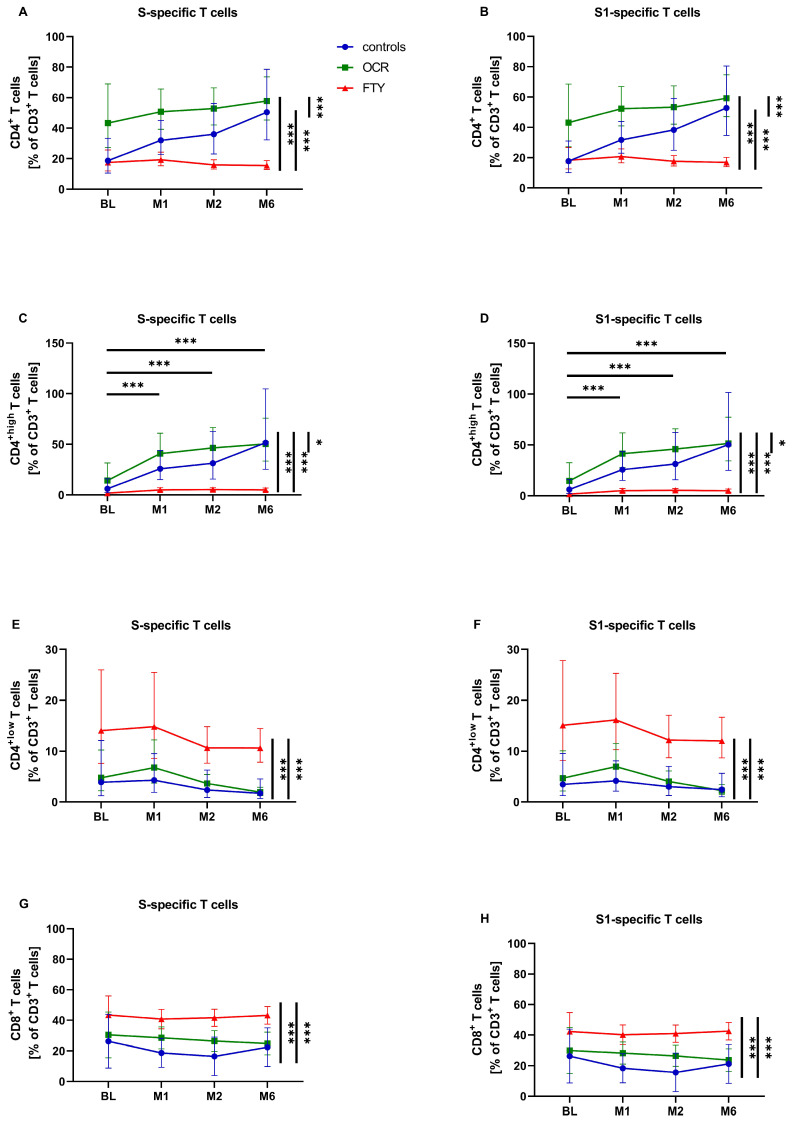
Impact of first SARS-CoV-2 mRNA vaccination on selected spike-specific (S) T-cell subsets and T-cell subsets specific for the S1 domain of the spike protein (S1) in the control group (controls, *n* = 17) compared to the fingolimod (FTY, *n* = 15) and the ocrelizumab (OCR, *n* = 15) groups. Relative percentages of T-cell subsets are shown at baseline (BL), one-month follow-up (M1), two-month follow-up (M2), and six-month follow-up (M6). Means with 95% confidence intervals are presented for relative percentages of CD4^+^ T cells (**A**,**B**), CD4^+high^ T cells (**C**,**D**), CD4^+low^ T cells (**E**,**F**), and CD8^+^ T cells (**G**,**H**). Asterisks indicate a statistically significant difference in percentages of relative T-cell subsets between selected time points or groups (* *p* < 0.05, *** *p* < 0.001).

**Figure 2 pathogens-14-00235-f002:**
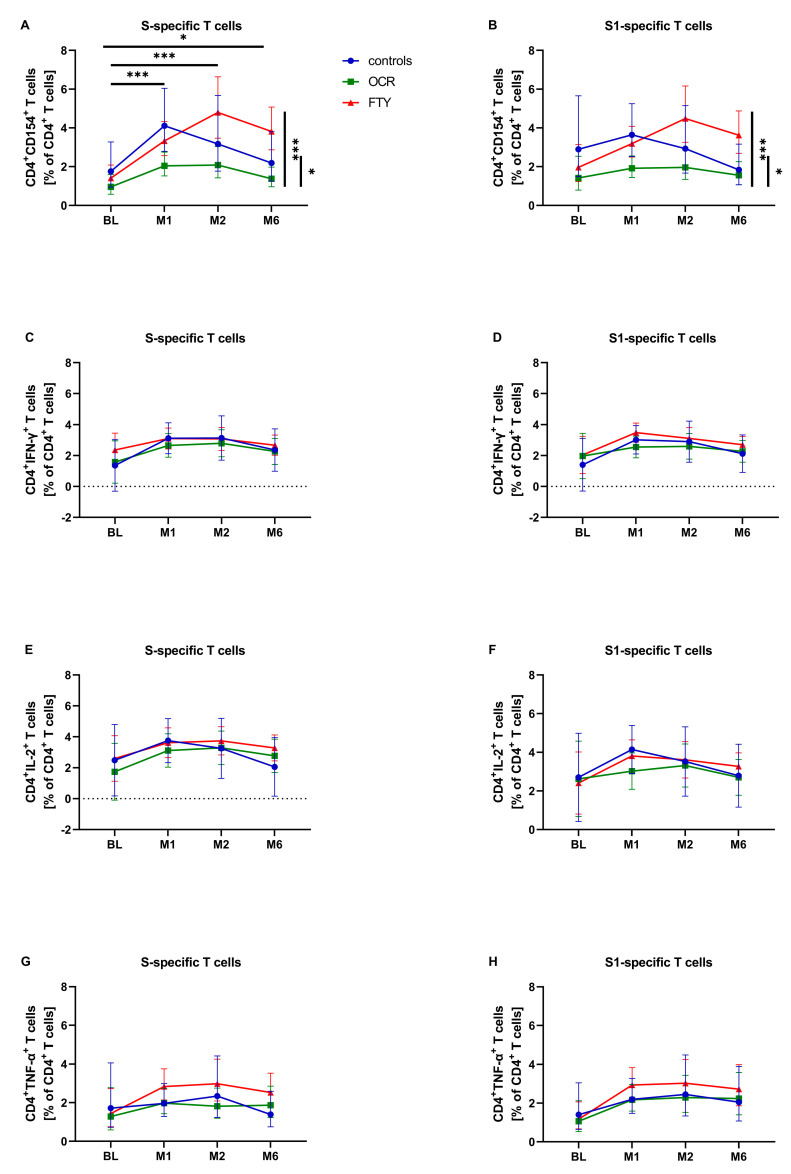
Impact of first SARS-CoV-2 mRNA vaccination on selected spike-specific (S) T-cell subsets and T-cell subsets specific for the S1 domain of the spike protein (S1) in the control group (controls, *n* = 17) compared to the fingolimod (FTY, *n* = 15) and the ocrelizumab (OCR, *n* = 15) groups. Relative percentages of T-cell subsets are shown at baseline (BL), one-month follow-up (M1), two-month follow-up (M2), and six-month follow-up (M6). Means with 95% confidence intervals are presented for relative percentages of CD4^+^CD154^+^ T cells (**A**,**B**), CD4^+^IFN-γ^+^ T cells (**C**,**D**), CD4^+^IL-2^+^ T cells **(E**,**F**), and CD4^+^TNF-α^+^ T cells (**G**,**H**). Asterisks indicate a statistically significant difference in relative percentages of T-cell subsets between selected time points or groups (* *p* < 0.05, *** *p* < 0.001).

**Figure 3 pathogens-14-00235-f003:**
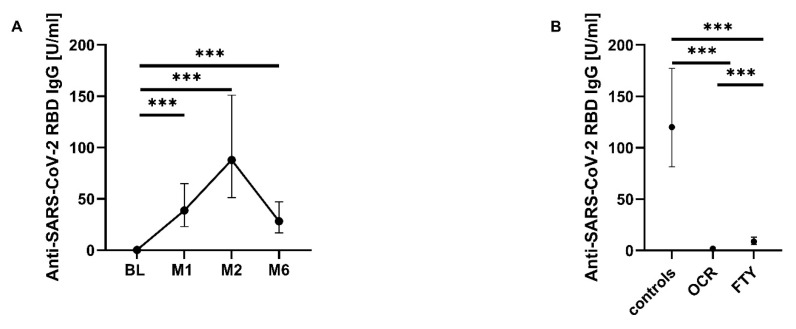
Impact of first SARS-CoV-2 mRNA vaccination on anti-SARS-CoV-2 RBD IgG titers. Means with 95% confidence intervals of anti-SARS-CoV-2 RBD IgG titers are shown. Anti-SARS-CoV-2 RBD IgG levels for the entire study cohort at different time points (**A**) and anti-SARS-CoV-2 RBD IgG levels overaged over all time points divided by patient groups (**B**). The lower detection limit was 0.43 U/mL, with values below this threshold reported as 0.2 U/mL. The upper detection limit was 25,000 U/mL. Asterisks indicate a statistically significant difference between selected time points or groups (*** *p* < 0.001). BL (baseline); M1 (one-month follow-up); M2 (two-month follow-up); M6 (six-month follow-up).

**Table 1 pathogens-14-00235-t001:** Demographics and clinical characteristics of patients.

Characteristic	All *n* = 47	Controls *n* = 17	Ocrelizumab *n* = 15	Fingolimod *n* = 15	*p*-Value
Age, years (mean, SD)	45.60 ± 11.89	48.18 ± 15.17	42.53 ± 10.61	45.73 ± 8.40	0.416
Sex, *n* (%) Female Male	35 (74.5) 12 (25.5)	13 (76.5) 4 (23.5)	14 (93.3) 1 (6.7)	8 (53.3) 7 (46.7)	0.041
Disease course, *n* (%) RRMS SPMS PPMS	38 (80.9) 4 (8.5) 5 (10.6)	12 (70.6) 3 (17.7) 2 (11.8)	12 (80.0) 0 (0.0) 3 (20.0)	14 (93.3) 1 (6.7) 0 (0.0)	
Disease duration, years (mean, SD)	13.47 ± 9.97	17.76 ± 11.85	10.20 ± 9.89	11.87 ± 5.66	0.054
Treatment duration, years (mean, SD) Alemtuzumab Natalizumab Ocrelizumab Fingolimod		7.83 ± 4.67 6.40 ± 2.19	2.07 ± 0.88	7.33 ± 2.92	
EDSS (mean, SD) (Median) (IQR)	3.47 ± 1.93 3.00 5.00	4.16 ± 2.20 3.75 4.25	2.97 ± 1.84 2.50 1.50	3.21 ± 1.57 2.75 4.00	0.270
Last infusion–first vaccination, days (mean, SD)			90.73 ± 42.69		
BL–first vaccination, days (mean, SD)	5.72 ± 10.44	5.24 ± 7.59	4.07 ± 10.87	7.93 ± 12.88	0.212
First vaccination–M1, days (mean, SD)	28.49 ± 6.27	29.00 ± 9.37	27.53 ± 3.09	28.87 ± 4.21	0.655
First vaccination–M2, days (mean, SD)	57.45 ± 7.08	56.47 ± 8.30	60.07 ± 7.49	55.93 ± 4.37	0.176
First vaccination–M6, days (mean, SD)	184.19 ± 18.18	187.76 ± 19.79	176.20 ± 20.20	188.13 ± 11.41	0.335

Values shown are those at baseline (exceptions: time intervals between last infusion and first vaccination, time intervals between first vaccination and measurement time points). *n* (patient count); SD (standard deviation); IQR (interquartile range); RRMS (relapsing–remitting multiple sclerosis); SPMS (secondary progressive multiple sclerosis); PPMS (primary progressive multiple sclerosis); EDSS (Expanded Disability Status Scale), baseline (BL = date of blood collection before first vaccination with SARS-CoV-2 messenger ribonucleic acid (mRNA) vaccine); M1 (one-month follow-up); M2 (two-month follow-up); M6 (six-month follow-up).

**Table 2 pathogens-14-00235-t002:** Vaccine-specific data in the patient groups.

		All *n* = 47	Controls *n* = 17	Ocrelizumab *n* = 15	Fingolimod *n* = 15
First-dose vaccine BNT162b2 mRNA-1273	*n* (%) *n* (%)	45 (95.7) 2 (4.3)	17 (100.0)	15 (100.0)	13 (86.7) 2 (13.3)
Second-dose vaccine BNT162b2 mRNA-1273	*n* (%) *n* (%)	45 (95.7) 2 (4.3)	17 (100.0)	15 (100.0)	13 (86.7) 2 (13.3)
Third-dose vaccine BNT162b2 mRNA-1273	*n* (%) *n* (%)	45 (95.7) 2 (4.3)	17 (100.0)	15 (100.0)	13 (86.7) 2 (13.3)
M1 1 vaccination 2 vaccinations	*n* (%) *n* (%)	17 (36.2) 30 (63.8)	4 (23.5) 13 (76.5)	8 (53.3) 7 (46.7)	5 (33.3) 10 (66.7)
M2 1 vaccination 2 vaccinations	*n* (%) *n* (%)	1 (2.1) 46 (97.9)	1 (5.9) 16 (94.1)	15 (100.0)	15 (100.0)
M6 2 vaccinations 3 vaccinations	*n* (%) *n* (%)	41 (87.2) 6 (12.8)	17 (100.0)	12 (80.0) 3 (20.0)	12 (80.0) 3 (20.0)

The vaccines used in the cohort and the number of vaccinations at each time point after BL are shown. M1 (one-month follow-up); M2 (two-month follow-up); M6 (six-month follow-up).

**Table 3 pathogens-14-00235-t003:** Relative percentages of SARS-CoV-2-specific T cells in relation to anti-SARS-CoV-2 RBD IgG titers.

T-Cell Subsets (%)	Patient Group	S		S1	
		IgG +	IgG −	IgG +	IgG −
CD4^+^ (% of CD3^+^)	All	28.622 CI (22.923, 35.376)	32.263 CI: (26.721, 38.953)	28.954 CI: (23.258, 36.046)	33.994 CI: (28.362, 40.745)
	Controls	25.378 CI: (18.798, 34.263)	41.148 CI: (24.126, 70.300)	24.855 CI: (18.567, 33.273)	42.875 CI: (25.716, 71.485)
	OCR	50.464 CI: (36.137, 70.701)	51.330 CI: (45.299, 58.163)	50.924 CI: (36.620, 70.816)	52.413 CI: (46.315, 59.315)
	FTY	18.278 (CI: 14.714, 22.706)	15.886 CI: (13.764, 18.355)	19.178 CI: (15.481, 23.759)	17.481 CI: (15.198, 20.107)
CD4^+high^ (% of CD3^+^)	All	11.042 * CI: (7.564, 16.119)	18.748 * CI: (13.925, 25.241)	11.219 CI: (7.710, 16.325)	18.654 CI: (13.891, 25.049)
	Controls	13.321 CI: (8.131, 21.825)	37.877 CI: (16.399, 87.487)	13.636 CI: (8.364, 22.230)	36.739 CI: (16.039, 84.158)
	OCR	27.456 CI: (15.749, 47.886)	42.377 CI: (34.560, 51.962)	28.239 CI: (16.263, 49.033)	42.429 CI: (34.585, 52.052)
	FTY	3.681 CI: (2.543, 5.327)	4.106 CI: (3.251, 5.185)	3.667 CI: (2.541, 5.290)	4.164 CI: (3.304, 5.247)
CD4^+low^ (% of CD3^+^)	All	4.802 CI: (3.269, 7.053)	5.544 CI: (3.632, 8.463)	4.834 CI: (3.331, 7.017)	6.646 CI: (4.631, 9.537)
	Controls	3.461 CI: (1.877, 6.381)	2.373 CI: (0.685, 8.224)	3.259 CI: (1.898, 5.598)	3.143 CI: (1.112, 8.880)
	OCR	2.306 CI: (1.271, 4.187) ***	6.499 CI: (5.129, 8.234) ***	2.328 CI: (1.294, 4.188) ***	7.368 CI: (5.876, 9.239) ***
	FTY	13.870 CI: (9.524, 20.199)	11.050 CI: (8.427, 14.488)	14.889 CI: (10.347, 21.426)	12.676 CI: (9.791, 16.411)
CD8^+^ (% of CD3^+^)	All	32.830 CI: (25.810, 39.850)	27.732 CI: (22.409, 33.055)	32.298 CI: (25.271, 39.326)	27.040 CI: (21.682, 32.397)
	Controls	23.661 CI: (14.653, 32.670)	18.289 CI: (3.387, 33.190)	23.584 CI: (14.549, 32.618)	17.214 CI: (2.204, 32.224)
	OCR	29.867 CI: (19.631, 40.103)	25.290 CI: (21.559, 29.020)	29.119 CI: (18.869, 39.369)	25.126 CI: (31.349, 28.904)
	FTY	44.962 CI: (38.099, 51.824)	39.618 CI: (35.354, 43.882)	44.192 CI: (37.323, 51.061)	38.779 CI: (34.520, 43.038)
CD4^+^CD154^+^ (% of CD4^+^)	All	1.690 ** CI: (1.313, 2.174)	3.182 ** CI: (2.528, 4.006)	2.060 CI: (1.558, 2.724)	2.924 CI: (2.337, 3.658)
	Controls	2.125CI: (1.519, 2.973)	3.329 CI: (1.788, 6.199)	2.746 CI: (1.932, 3.902)	2.743 CI: (1.514, 4.970)
	OCR	1.118 CI: (0.721, 1.732)	2.122 CI: (1.796, 2.508)	1.389 CI: (0.883, 2.185)	2.070 CI: (1.741, 2.461)
	FTY	2.031 CI: (1.583, 2.605)	4.562 CI: (3.728, 5.582)	2.292 CI: (1.740, 3.019)	4.404 CI: (3.592, 5.399)
CD4^+^IFN-γ^+^ (% of CD4^+^)	All	2.202 CI: (1.539, 2.865)	2.882 CI: (2.303, 3.460)	2.282 CI: (1.593, 2.972)	2.751 CI: (2.220, 3.281)
	Controls	1.851 CI: (0.969, 2.734)	3.137 CI: (1.544, 4.730)	1.886 CI: (1.007, 2.766)	2.839 CI: (1.378, 4.300)
	OCR	2.263 CI: (1.275, 3.451)	2.288 CI: (1.877, 2.698)	2.556 CI: (1.518, 3.595)	2.143 CI: (1.753, 2.532)
	FTY	2.390 CI: (1.740, 3.041)	3.220 CI: (2.735, 3.705)	2.404 CI: (1.728, 3.080)	3.270 CI: (2.824, 3.715)
CD4^+^IL-2^+^ (% of CD4^+^)	All	2.882 CI: (1.991, 3.773)	3.068 CI: (2.265, 3.872)	3.037 CI: (2.119, 3.956)	3.284 CI: (2.569, 4.000)
	Controls	3.029 CI: (1.806, 4.252)	2.745 CI: (0.489, 5.002)	3.034 CI: (1.855, 4.213)	3.546 CI: (1.572, 5.521)
	OCR	2.638 CI: (1.201, 4.075)	2.819 CI: (2.277, 3.361)	3.151 CI: (1.769, 4.533)	2.683 CI: (2.164, 3.203)
	FTY	2.978 CI: (2.107, 3.850)	3.641 CI: (3.006, 4.276)	2.927 CI: (2.026, 3.828)	3.623 CI: (3.030, 4.217)
CD4^+^TNF-α^+^ (% of CD4^+^)	All	1.815 CI: (1.262, 2.609)	2.083 CI: (1.613, 2.690)	1.800 CI: (1.294, 2.505)	2.311 CI: (1.798, 2.969)
	Controls	2.146 CI: (1.381, 3.335)	1.545 CI: (0.783, 3.049)	1.964 CI: (1.310, 2.944)	2.001 CI: (1.040, 3.849)
	OCR	1.484 CI: (0.861, 2.560)	1.980 CI: (1.621, 2.419)	1.723 CI: (1.021, 2.909)	1.995 CI: (1.635, 2.434)
	FTY	1.875 CI: (1.313, 2.679)	2.955 CI: (2.336, 3.737)	1.725 CI: (1.245, 2.390)	3.091 CI: (2.440, 3.914)

Impact of first SARS-CoV-2 mRNA vaccination on selected spike (S) specific T-cell subsets and T-cell subsets specific for the S1 domain of the spike protein (S1) as a function of the serostatus of anti-SARS-CoV-2 receptor-binding domain (RBD) immunoglobulin G (IgG) titers. Relative percentages of T-cell subsets are shown for seropositive (IgG +) compared to seronegative (IgG −) individuals. Means with 95% confidence intervals are presented. OCR, ocrelizumab; FTY, fingolimod. Asterisks indicate a statistically significant difference in relative percentages of T-cell subsets based on the serostatus in the corresponding patient group (* *p* < 0.05, ** *p* < 0.01, *** *p* < 0.001).

**Table 4 pathogens-14-00235-t004:** Serostatus of anti-SARS-CoV-2 RBD IgG over time in patient groups.

		BL	M1	M2	M6
All seronegative seropositive	*n* (%) *n* (%)	45 (97.8) 1 (2.2)	22 (46.8) 25 (53.2)	18 (38.3) 29 (61.7)	18 (38.3) 29 (61.7)
Controls seronegative seropositive	*n* (%) *n* (%)	17 (100.0)	3 (17.6) 14 (82.4)	17 (100.0)	17 (100.0)
Ocrelizumab seronegative seropositive	*n* (%) *n* (%)	15 (100.0)	11 (73.3) 4 (26.7)	12 (80.0) 3 (20.0)	13 (86.7) 2 (13.3)
Fingolimod seronegative seropositive	*n* (%) *n* (%)	13 (92.9) 1 (7.1)	8 (53.3) 7 (46.7)	6 (40.0) 9 (60.0)	5 (33.3) 10 (66.7)

The serostatus of anti-SARS-CoV-2 RBD IgG is presented for the overall study cohort and each patient group across all time points. BL (baseline); M1 (one-month follow-up); M2 (two-month follow-up); M6 (six-month follow-up).

## Data Availability

Dataset will be made available by the authors upon request.

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
