# Peer review of "Extensive T-Cell Profiling Following SARS-CoV-2 mRNA Vaccination in Multiple Sclerosis Patients Treated with DMTs"

_pathogens, 2025, doi:10.3390/pathogens14030235_

Round 1
Reviewer 1 Report
Comments and Suggestions for Authors
Your paper provides helpful information on the cellular responses to SARS-CoV-2 mRNA vaccination in patients with MS, in this case treated with anti-CD20 and S1P1R modulator agents. You excluded or used as control patients on natalizumab and alemtuzumab in view their MOA would not have any influence in the immune responses and vaccination. The question arises if you did not have enough people on cladribine (a more vigorous immunosuppressor) to be studied, or if there is another rationale for such exclusion.
Author Response
Dear Reviewer,
Thank you very much for the critical review of our manuscript.
Please find attached our response to your comments and the revised version of our manuscript.
Best regards, Hannah Solchenberger and Katja Akgün
Comment:
Your paper provides helpful information on the cellular responses to SARS-CoV-2 mRNA vaccination in patients with MS, in this case treated with anti-CD20 and S1P1R modulator agents. You excluded or used as control patients on natalizumab and alemtuzumab in view their MOA would not have any influence in the immune responses and vaccination. The question arises if you did not have enough people on cladribine (a more vigorous immunosuppressor) to be studied, or if there is another rationale for such exclusion.
Response:
Thank you for your review. While conducting our study during the Covid-19 pandemic, there was an insufficient number of PwMS at our center who had undergone an intensive immunosuppressive treatment cycle with agents such as cladribine or alemtuzumab early before SARS-CoV-2 mRNA vaccination. Due to the strong immunosuppressive effect, these therapies were explicitly avoided during the pandemic because of the increased risk of severe disease. We have included a statement in the methodology section clarifying that the number of PwMS treated with potent immunosuppressive drugs was insufficient for analysis during the study period.
Reviewer 2 Report
Comments and Suggestions for Authors
The authors present an exhaustive study on the profiling of T cells in MS patients following vaccination with SARS-CoV-2 mRNA vaccines. The main conclusions are that MS treatment with anti-CD20 or S1PR did not inhibit a broad CD4+T cell response and that humoral response was suppressed in treated MS patients. Their findings demonstrate a compensatory upregulation of SARS-CoV-2-specific CD4+ T cells in the absence of seroconversion in pwMS.
This is a well-performed study that contributes some new understanding in cellular immunity of SARS-CoV-2-vaccinated MS patients. A criticism could be the relatively limited size of the compared groups, and lack of validation in an independent cohort. The article is generally well written and easy to follow, but specifically the Discussion is very long and misses a bit of focus. Limitations of the study are suitably addressed in the Discussion.
I find this an interesting and well-performed study.
Minor comments:
- Table 2: define M1, M2 and M6 in footnote
- Specifics of the BNT162b2 and mRNA-1273 vaccines need to be defined in the Materials and Methods (e.g. supplier, dose, to which SARS-CoV-2 epitope is it directed?, etc.)
- The anti-SARS-CoV-2 RBD test needs to be described in the Materials and Methods
- Line 207; please define what are CD4+high and CD4+low cells. It would be useful to provide some context of the relevance of this distinction in regard of this study and to provide some references. This could be done in the introduction.
- In Figure 2 and accompanying text S-specific and S1-specific T cells are highlighted.Some explanation on what the difference is between both epitopes would be useful.
Author Response
Dear Reviewer,
Thank you very much for the critical review of our manuscript.
Please find attached our response to your comments and the revised version of our manuscript.
Best regards, Hannah Solchenberger and Katja Akgün
Comment:
A criticism could be the relatively limited size of the compared groups, and lack of validation in an independent cohort.
Response:
Thank you for your review. The points of criticism have been incorporated into the limitations section of the discussion.
Comment:
The article is generally well written and easy to follow, but specifically the Discussion is very long and misses a bit of focus.
Response:
We have shortened some parts of the discussion.
Comment:
Table 2: define M1, M2 and M6 in footnote.
Response:
We defined M1, M2 and M6 in the footnote of all corresponding tables and figures.
Comment:
Specifics of the BNT162b2 and mRNA-1273 vaccines need to be defined in the Materials and Methods (e.g. supplier, dose, to which SARS-CoV-2 epitope is it directed?, etc.)
Response:
We defined specifics of the BNT162b2 and mRNA-1273 vaccine in the methods section.
Comment:
The anti-SARS-CoV-2 RBD test needs to be described in the Materials and Methods
Response:
The anti-SARS-CoV-2 RBD test was performed in a certified laboratory affiliated with the Institute of Transfusion Medicine, Dresden, Germany. An automated immunoassay system (cobas e 801) was used to perform the electrochemiluminescence immunoassay for the in vitro qualitative detection of antibodies against SARS-CoV-2. We apologize if it was not clear that this is an automated procedure carried out under standardized conditions. We have added a this aspect for clarification in the methodology section.
Comment:
Line 207; please define what are CD4+high and CD4+low cells. It would be useful to provide some context of the relevance of this distinction in regard of this study and to provide some references. This could be done in the introduction.
Response:
We highlighted that CD4+ T cells are divided into subpopulations of CD4+high and CD4+low T cells based on their fluorescence intensity. In addition, we referred to a previous study demonstrating that CD4+ and CD8+ T cells can be distinguished into subpopulations based on their antigen expression, showing either high or low fluorescence intensity after staining with monoclonal anti-CD4 or anti-CD8 antibodies.
Comment:
In Figure 2 and accompanying text S-specific and S1-specific T cells are highlighted. Some explanation on what the difference is between both epitopes would be useful.
Response:
We have added a sentence in the methodology section explaining the difference between S-and S1-specific T cells and emphasized that they were added separately to the T cells.
Reviewer 3 Report
Comments and Suggestions for Authors
This research investigates the immune response to SARS-CoV-2 mRNA vaccines in multiple sclerosis patients, a population often treated with disease-modifying therapies (DMTs). Focusing on patients receiving ocrelizumab or fingolimod, the study demonstrates that while humoral immunity can be compromised by these DMTs, a CD4+ T cell response is still elicited by vaccination. This finding points to a potential compensatory role for T cell immunity in the context of impaired B cell responses, underscoring the necessity of evaluating both arms of the immune system when assessing vaccine efficacy in vulnerable populations.
- Please expand the description of the flow cytometry assay to include more specifics ,eg : catalogue number and dilutions of antibody, what does the percentage mean between line 107-112, (weight/volume (w/v)?, for the lyophilized peptides or proteins). Providing more comprehensive information will enhance the reader's understanding and facilitate the application of this assay in future studies. The authors only showed the percentage of T cells, but did not include cell count data, despite mentioning it in the text. Please add figures of such data to the manuscript. Additionally, the summary figure legends for Figures 1, 2, and S2 are nearly identical and should be more specific. They are described as showing the "Impact of the first SARS-CoV-2 mRNA vaccination." However, at months 2, 3, and 6, some patients have received 2 or 3 doses of vaccines. The current description is confusing or inaccurate.
- The statistics labels are confusing, making it hard to determine what comparisons between groups are being made. Subpanels within Figure 1 (A and B, C and D, E and F) appear very similar. The methods section does not clearly describe the specific differences between the stimulations (S-specific and S1-specific, were they added together or individually?) highlighted in these paired subpanels. Please explicitly state in the methods.
- For consistency and ease of comparison with Figures 1 and 2, consider restructuring Figure 3 to present data by time points and treatment groups. Furthermore, clearly indicate and label the assay's detection limit on Figure 3.
Author Response
Dear Reviewer,
Thank you very much for the critical review of our manuscript.
Please find attached our response to your comments and the revised version of our manuscript.
Best regards, Hannah Solchenberger and Katja Akgün
Comment:
Please expand the description of the flow cytometry assay to include more specifics ,eg : catalogue number and dilutions of antibody, what does the percentage mean between line 107-112, (weight/volume (w/v)?, for the lyophilized peptides or proteins). Providing more comprehensive information will enhance the reader's understanding and facilitate the application of this assay in future studies. The authors only showed the percentage of T cells, but did not include cell count data, despite mentioning it in the text. Please add figures of such data to the manuscript. Additionally, the summary figure legends for Figures 1, 2, and S2 are nearly identical and should be more specific. They are described as showing the "Impact of the first SARS-CoV-2 mRNA vaccination." However, at months 2, 3, and 6, some patients have received 2 or 3 doses of vaccines. The current description is confusing or inaccurate.
Response:
Thank you for your review.
We have added detailed information on the reagents, chemicals, solutions, buffers, fluorescent-labeled antibodies, and beads used in the present study to the Supplementary Material section to facilitate the application of this assay in future studies.
Regarding the percentage of T cells, we apologize for using the term “T cell counts”. In our study, we solely relied on relative percentages of T cells for our analyses and have updated the wording in the study accordingly.
We have updated the legends of the figures and tables to make them more specific and clearer. We have also revised the terminology in the legends of the tables and figures as well as in the text, changing “one, two, and six months after vaccination” to “one-month follow-up”, “two-month follow-up”, and “six-month follow-up” to avoid any misunderstanding. As mentioned in limitation section of the discussion, we did not specifically analyze the immune response following the second vaccination.
Comment:
The statistics labels are confusing, making it hard to determine what comparisons between groups are being made. Subpanels within Figure 1 (A and B, C and D, E and F) appear very similar. The methods section does not clearly describe the specific differences between the stimulations (S-specific and S1-specific, were they added together or individually?) highlighted in these paired subpanels. Please explicitly state in the methods.
Response:
Regarding the statistical labeling, given the numerous differences between the three patient groups in our study, it was challenging to present all the necessary information clearly. Therefore, we added a table in the Supplementary Material section (Table S5) that explicitly shows the significant and non-significant effects between patient groups for Figures 1 and 2. We have added a sentence in the methods section explaining the difference between S-and S1-specific T cells and emphasized that they were added separately to the T cells.
Comment:
For consistency and ease of comparison with Figures 1 and 2, consider restructuring Figure 3 to present data by time points and treatment groups. Furthermore, clearly indicate and label the assay's detection limit on Figure 3.
Response:
We considered a representation based on time points and treatment groups in Figure 3. However, the data could not be clearly displayed in this way, so we opted for the current representation. We have added the lower and upper detection limits of the assay in the legend of Figure 3 and included a note on the upper detection limit in the Methods section.